# The burden of premature mortality from cardiovascular diseases: A systematic review of years of life lost

Wan Shakira Rodzlan Hasani[1,2]* , Nor Asiah Muhamad[3] , Tengku Muhammad Hanis[1], Nur Hasnah Maamor[3], Chen Xin Wee[4], Mohd Azahadi Omar[5], Shubash Shander Ganapathy[2], Zulkarnain Abdul Karim[6], Kamarul Imran Musa[1]

1 Department of Community Medicine, School of Medical Sciences, Universiti Sains Malaysia, Kubang Kerian, Kelantan, Malaysia, 2 Institute for Public Health, National Institutes of Health, Ministry of Health Malaysia, Setia Alam, Selangor, Malaysia, 3 Sector for Evidence-Based Healthcare, National Institutes of Health, Ministry of Health Malaysia, Setia Alam, Selangor, Malaysia, 4 Department of Public Health Medicine, Faculty of Medicine, Sungai Buloh Campus, Universiti Teknologi MARA, Sungai Buloh, Selangor, Malaysia, 5 Sector for Biostatistics and Data Repository, National Institutes of Health, Ministry of Health Malaysia, Setia Alam, Selangor, Malaysia, 6 Office of The Manager, National Institutes of Health, Ministry of Health Malaysia, Setia Alam, Selangor, Malaysia

☸ These authors contributed equally to this work.

* shaki_iera@yahoo.com

**Data Availability Statement:** Data are available at https://github.com/shakirarodzlan/SR_ PrematureMortality.git.

## Abstract

### Introduction

Premature mortality refers to deaths that occur before the expected age of death in a given population. Years of life lost (YLL) is a standard parameter that is frequently used to quantify some component of an "avoidable" mortality burden.

### Objective

To identify the studies on premature cardiovascular disease (CVD) mortality and synthesise their findings on YLL based on the regional area, main CVD types, sex, and study time.

### Method

We conducted a systematic review of published CVD mortality studies that reported YLL as an indicator for premature mortality measurement. A literature search for eligible studies was conducted in five electronic databases: PubMed, Scopus, Web of Science (WoS), and the Cochrane Central Register of Controlled Trials (CENTRAL). The Newcastle-Ottawa Scale was used to assess the quality of the included studies. The synthesis of YLL was grouped into years of potential life lost (YPLL) and standard expected years of life lost (SEYLL) using descriptive analysis. These subgroups were further divided into WHO (World Health Organization) regions, study time, CVD type, and sex to reduce the effect of hetero-geneity between studies.

**Funding:** The authors received no specific funding for this work.

**Competing interests:** The authors have declared that no competing interests exist.

## Results

Forty studies met the inclusion criteria for this review. Of these, 17 studies reported premature CVD mortality using YPLL, and the remaining 23 studies calculated SEYLL. The selected studies represent all WHO regions except for the Eastern Mediterranean. The overall median YPLL and SEYLL rates per 100,000 population were 594.2 and 1357.0, respectively. The YPLL rate and SEYLL rate demonstrated low levels in high-income countries, including Switzerland, Belgium, Spain, Slovenia, the USA, and South Korea, and a high rate in middle-income countries (including Brazil, India, South Africa, and Serbia). Over the past three decades (1990–2022), there has been a slight increase in the YPLL rate and the SEYLL rate for overall CVD and ischemic heart disease but a slight decrease in the SEYLL rate for cerebrovascular disease. The SEYLL rate for overall CVD demonstrated a notable increase in the Western Pacific region, while the European region has experienced a decline and the American region has nearly reached a plateau. In regard to sex, the male showed a higher median YPLL rate and median SEYLL rate than the female, where the rate in males substantially increased after three decades.

## Conclusion

Estimates from both the YPLL and SEYLL indicators indicate that premature CVD mortality continues to be a major burden for middle-income countries. The pattern of the YLL rate does not appear to have lessened over the past three decades, particularly for men. It is vitally necessary to develop and execute strategies and activities to lessen this mortality gap.

## Systematic review registration

PROSPERO CRD42021288415

## Introduction

Cardiovascular disease (CVD), principally ischemic heart disease (IHD) and cerebrovascular disease, remains the major cause of premature mortality, accounting for about one-third of all deaths globally [1, 2], and this figure is predicted to escalate [3]. Moreover, the hit of the COVID-19 pandemic has refrained many countries from financing strategies to achieve Sustainable Development Goal (SDG) target 3.4 to reduce premature mortality from non-communicable diseases (NCDs) by 25% by 2025 [4]. Providing current data and information on geographic and sex differences in premature mortality may help global health players and payers develop context-specific strategies and provide relevant financial assistance in funding CVD interventions.

By definition, premature mortality is referred to as a death that occurs before the expected age of death in a given population [5]. Premature mortality is a very common population health indicator that is frequently used, for example, in international and national performance assessments, to measure some component of an "avoidable" burden of mortality. There are several methods to calculate the burden of premature mortality; (i) proportion of premature mortality under a chosen age threshold; (ii) age-standardized mortality rates under a defined age range; (iii) years of life lost (YLL) [6]; (iv) probability of dying between an exact age range, determined from the life table method [7]; and (v) standardized mortality ratio

(SMR), comparing the premature mortality of a study population to that of a reference population [8]

YLL is a standard parameter and a more accurate measurement of premature mortality. This indicator accounts for death numbers and the age at which the death occurs, giving more weight to deaths at younger ages [6, 9]. The method of calculating YLL varies from author to author. In general, two methods are commonly used to calculate YLL: i) years of potential life lost (YPLL) and ii) standard expected years of life lost (SEYLL). YPLL was first introduced in 1941 for the tuberculosis mortality study [10]. In 1971, Romeder et al. [9] refined the method of calculating YPLL as a useful mortality index for health planning, and then in 1990, the formula for YPLL was adopted by Gardner [6]. YPLL was commonly used because it was easy to calculate by subtracting the age of death from a chosen cut-off (e.g., 65, 75, or 85 years) [6, 11]. The conventional age threshold measures of YPLL, however, do not account for deaths after the cut-off age, leading them to fail in capturing avoidable deaths at ages outside the selected age range. Furthermore, the selection of the upper age limit varies from study to study. In 1996, the Global Burden of Disease (GBD) study introduced SEYLL to address the issue of arbitrary age threshold selection [12]. The SEYLL formula is based on comparing the age of death to the standard life expectancy of an individual at each age and incorporates time discounting and age weighting. Consequently, SEYLL is increasingly used as an indicator of premature mortality to calculate the mortality-associated disease burden.

Despite a growing number of individual studies reporting YLL as an indicator of premature mortality, the authors' limited search revealed no recent review synthesis focusing on CVD-related premature death using YLL. Several systematic reviews, but not specifically on CVD deaths, have been conducted to investigate premature mortality using YLL [13–15]. Therefore, we conducted a systematic review to identify studies and synthesise their findings on YLL from CVD based on the method from the Gardner and GBD study. We aimed to stratify the findings by region, sex, main CVD types, and study time.

## Method

We followed the Preferred Reporting Items for Systematic Reviews and Meta-analysis (PRISMA) for this review [16] (**S1 Checklist**). The protocol of this review was registered in the International Prospective Register of Systematic Reviews (PROSPERO), systematic review registry (CRD42021288415).

### Search strategy

We searched the electronic databases of PubMed, Scopus, Web of Science (WoS), and the Cochrane Central Register of Controlled Trials (CENTRAL) to identify eligible studies. We cross checked all eligible articles from the reference list of included articles. We searched Google Scholar to identify articles that were not indexed in the major electronic databases. All databases were searched from their inception through October 18, 2022. Our search strategy included terms for "cardiovascular diseases" (e.g., coronary heart disease, cerebrovascular disorder, myocardial ischemia, or stroke) and the term for premature mortality (e.g., premature death, years of life lost, potential years of life lost, and standard expected year of life lost). The search was limited for studies in English-language only. The detail search terms for each database is presented in **S1 Table**.

### Study selection

The Mendeley Reference Management Software (https://www.mendeley.com) was used to store, organize, and manage all the references. Prior to the screening process, all the search

**Table 1. Formulas for years of potential life lost (YPLL) and standard expected years of life lost (SEYLL).**

| | YPLL | SEYLL |
|---|---|---|
| Formula proposed by; | Gardner (1990) [6] and Romeder (1977) [9] | Global Burden of Disease (1996) [12] |
| Formula for total number YLL | $$Total\ YPLL = \sum_{i=0}^{N} di\,(N-i)$$ | $$Total\ SEYLL = \sum_{x=0}^{l} d_x\,e_x^*$$ |
| | Where, $i$ is age at death, $di$ is number of deaths at age $i$, and N is upper cut-off age. | Where, $d_x$ is the number of deaths and $e_x^*$ is expected years of life at each age in the standard population |
| Formula for YLL rate | $YPLL\ rate = \frac{Total\ YPLL}{N} X\ 100,000$ | $SEYLL\ rate = \frac{Total\ SEYLL}{N} X\ 100,000$ |
| | Where, N is total reference population | Where N is the number of people at x age |
| Formula for YLL per death | $YPLL\ per\ death = \frac{Total\ YPLL}{Number\ of\ CVD\ deaths}$ | $SEYLL\ per\ death = \frac{Total\ SEYLL}{Number\ of\ CVD\ deaths}$ |

results were imported into Mendeley, and a duplicate paper was deleted by one author (W.S.R. H). We divided the screening process into two phases. For the first phase, four authors (W.S.R. H, H.M, C.X.W and N.A.M) independently screened the titles and abstracts to examine the potential studies for inclusion and exclude those that were obviously irrelevant. Studies were included if they (1) reported premature mortality due to CVD, (2) used an observational study design, and (3) were written in English. We excluded reviews, meta-analyses, letters, comments, and editorials.

We retrieved the full-text articles for the potentially relevant studies in the second phase. Two review authors (W.S.R.H and T.M.H) independently screened the full-text articles and identified studies for inclusion according to the eligibility criteria, and recorded the reasons for exclusion of the excluded studies. We resolved any disagreements (phases 1 and 2) through discussion or, whenever necessary, we consulted a third review author (N.A.M). If no consensus could be reached, another author (K.I.M) would act as an arbiter. We recorded the selection process and completed the Preferred Reporting Items for Systematic Reviews and Meta-Analyses (PRISMA) flow diagram [16]. We included articles that reported YLL in rate or YLL per death as an indicator of premature CVD mortality. Any terms for YLL, including YPLL, premature years of potential life lost (PYLL), age-standardized YLL rates (ASYR), or SEYLL, were included as long as they used the method based on Gardner (1990) [6], Romeder (1977) [9], or the GBD study (1996) [12] to calculate YLL. For YPLL (formula by Gardner or Romeder), any upper age limit (e.g., < 70 or < 65) that was defined by the study as premature mortality was included in this review. Meanwhile, the SEYLL formula from GBD requires the standard number of expected years of life for each age group. Therefore, any standard life expectancy used by studies to calculate SEYLL was accepted. For the YLL rate, due to the difference in the denominator between per person and per person at risk (or population at risk), we decided to include the YLL rate per person or population (e.g., YLL rate per 1,000, 10,000, or 100,000 population). The details of the formulas for YPLL and SEYLL are presented in **Table 1**. We excluded studies that only reported the absolute number of YLL (with no information on the number of CVD deaths, the YLL rate, or YLL per death) or that were restricted to a specific population or very specific medical condition (e.g., epilepsy or congenital disease).

## Data extraction and management

Two review authors (W.S.R.H and T.M.H) independently extracted the data according to guidance from the Cochrane Handbook for Systematic Reviews of Interventions. We used a standard data extraction form created by the Microsoft Excel spreadsheet (**S2 Table**) for study characteristics and outcome data. One reviewer (W.S.R.H) conducted a full abstraction of all

data, and another reviewer (T.M.H) verified for the accuracy. From all eligible articles, we abstracted the first author's name, year of publication, country, data setting, year of data collected, study design, study population, data source, number of deaths, types of CVD death, method or formula used for YLL calculation, and YLL value, including total YLL, YLL rate, and YLL per death. For each type of YLL measure (YPLL and SEYLL), the YLL value was abstracted separately based on CVD types, sex, or age. We contacted the author to get the exact value for YLL if they reported it in the plot or reported YLL as part of all-cause or general NCD mortality.

We separated all results according to the method of calculation for YLL, which were YPLL and SEYLL. Due to the various terms used by the authors to report YLL, we standardised the terms as YPLL for any study that used Gardner's method or Romeder's method and SEYLL for studies that used the GBD method. The main outcomes in this review were the YLL rate per 100,000 and the YLL per death due to CVD. For the YPLL and SEYLL rates, we used the rate per 100,000 persons as the standard value for this review by converting other rates (e.g., per 1,000 or per 10,000) into 100,000. If the study did not report the YPLL or SEYLL per death, we calculated it by dividing the total number of YLL by the total number of deaths, whenever data was available. We could not proceed with the meta-analysis as most of the studies did not report the numbers of CVD deaths, and none of them reported measures of uncertainty, including the 95% CI or standard error of the YLL rate or YLL per death. We summarised the results using a descriptive analysis rather than a meta-analysis. Each study may report YLL values (YPLL or SEYLL) for the overall population or may report the value for each sex or CVD type. Thus, we treat each value as separate data from each study to calculate the median and IQR (interquartile range). The YLL rate and YLL per death were summarised and presented in tables using median and range stratified by study time, the WHO regions, sex, and CVD type (all CVD, IHD, and cerebrovascular disease). We plotted the median YLL rates (YPLL and SEYLL) for each country that reported the data. We also plotted the pattern of the YPLL rate and SEYLL rate for the past three decades (1990–2022) based on CVD types, sex, and WHO regions. Data from some countries might have limited quality and representativeness. Hence, in the analysis, we exerted two assumptions: a) the data from each source represents the national population, and b) the measurement of the data was valid for all data sources.

## Quality assessment

The Newcastle-Ottawa Scale (NOS) criteria were used to assess the study quality for each included articles [17]. NOS applied a "star system," where the study is assessed based on three broad perspectives: 1) the selection of the study groups; 2) the comparability of the groups, and 3) the ascertainment of exposure/outcome. The original version of NOS was based on a cohort study and a case-control study design. We used the adapted NOS version by Herzog et al. [18] for cross-sectional study design, where they assess the same three components (selection, comparability, and outcome) as the original version. The score for the adapted version for the cross-sectional studies is as follows: 1) very good studies: 9–10 points; 2) good studies: 7–8 points; 3) satisfactory studies: 5–6 points; and 4) unsatisfactory studies: 0–4 points. The detailed criteria for NOS assessment are represented in **S3 Table**.

## Ethics and dissemination

This study was approved by the National Medical Research Register (NMRR), Ministry of Health Malaysia (NMRR ID-22-00231-MOX) and the Human Research Ethics Committee of USM (USM/JEPeM/22030181). There will be no concerns about privacy.

## Results

A total of 2012 studies were identified through the database search, and 30 additional studies were identified through the screening of reference lists. **Fig 1** illustrates the flow of information through the identification and screening phases of systematic review. There were 1291 studies screened for eligibility through title/abstract, subsequently through full-text, and finally yielding a total of 40 studies to be included for review synthesis. The NOS adapted version for cross-sectional design yielded 37 studies of good quality and three of satisfactory quality. None of the selected studies was of poor quality or unsatisfactory. Thus, we included all 40 studies in this review. The detailed quality assessment of each study was presented in **Table 2**.

### Characteristic of included studies

The world map (**Fig 2**) demonstrated the distribution of selected studies based on country level. Regardless of the method they used for YLL, the selected studies represented all WHO regions, including the Americas (9 studies), Europe (17 studies), South-East Asia (2 studies), the Western Pacific (11 studies), and Africa (1 study) except for the Eastern Mediterranean region. The characteristics of the included studies are listed in **Table 2**. Of the 40 studies, 17 reported premature CVD mortality using the YPLL method from Gardner (1990) or Romeder (1977), and the remaining 23 studies calculated SEYLL as proposed by GBD studies. The study years (time data source) ranged from 1969 to 2014 for YPLL and 1990 to 2017 for SEYLL. All the studies were of a cross-sectional design, and the majority of them used vital registration data, such as country mortality databases or censuses, as data sources. Thus, the majority of the studies were nationally representative, covering the general population of their country in that particular year of study. In terms of age coverage, it depends on the method used to

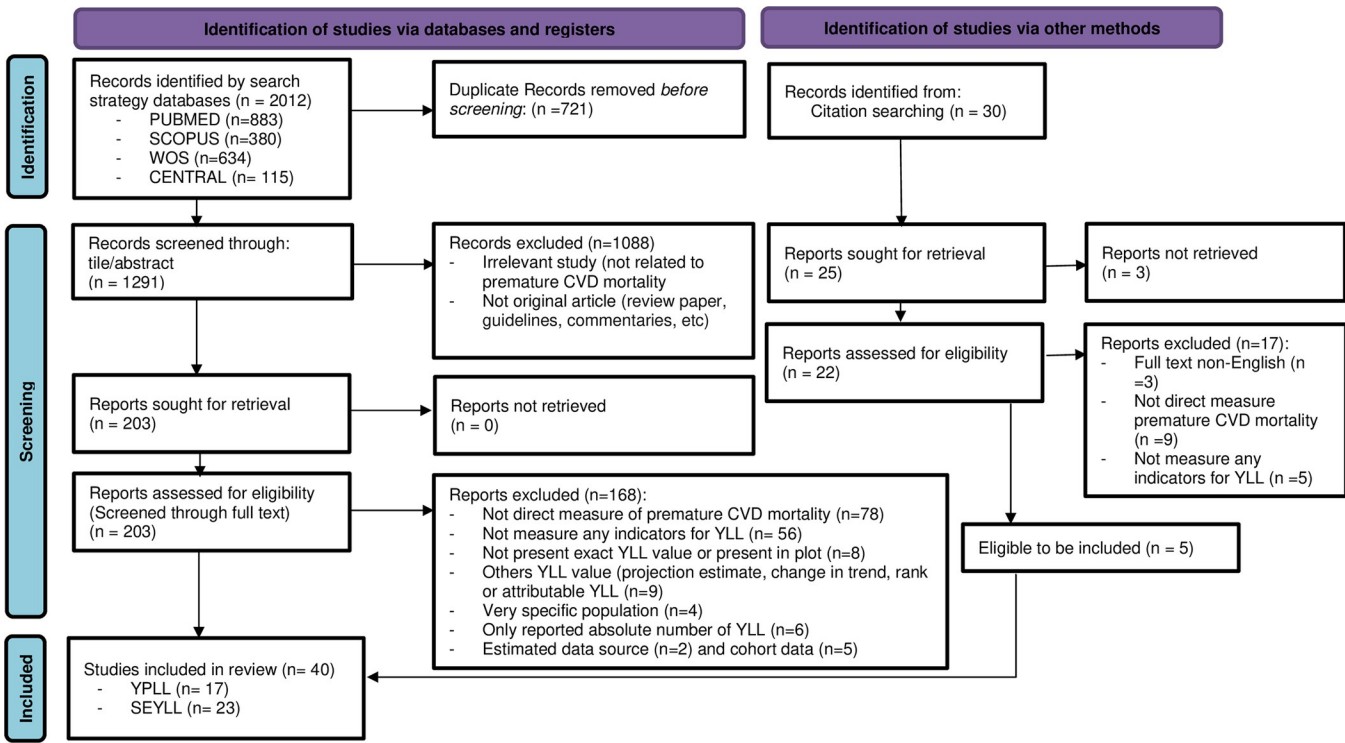

**Fig 1. Flow diagram of the published articles evaluated for inclusion in this review.**

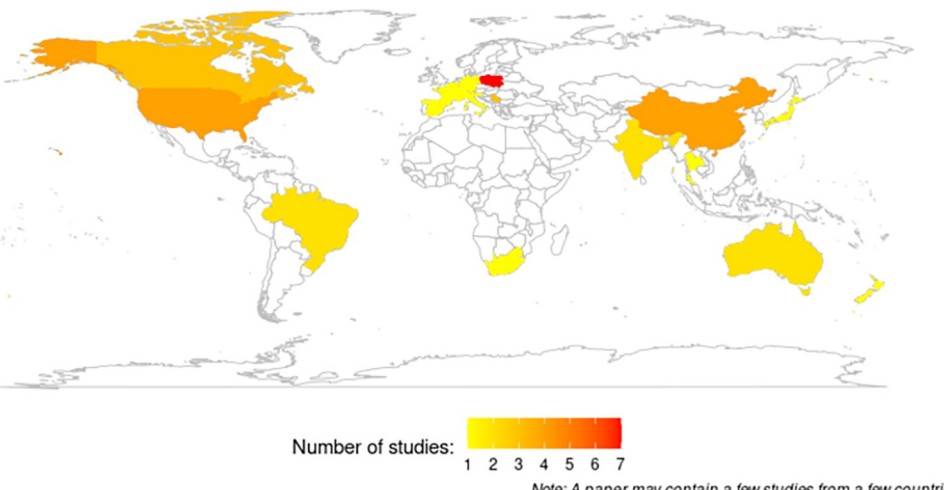

**Fig 2. Distribution of selected studies by country.**

calculate YLL. The studies that used the GBD method included all the age groups in the analysis as indicated by its formula. While the studies based on the Gardner or Romeder method needed upper age limit coverage according to their definition of premature mortality (e.g., less than 65, 70, or 75 years). Some studies started at the age of young adults (>15 or >20 years) and middle-aged adults (>30 or >40 years). In order to address the issue of this heterogeneity between studies, the results were separated and evaluated on the basis of the methods used for YLL calculation, which were YPLL and SEYLL. We also conducted subgroup analysis by regional area, CVD types, sex, and study time for each YPLL and SEYLL indicator.

## Years of potential life lost (YPLL) from CVD

**Table 3** shows the summary of premature CVD mortality studies using YPLL. Twelve studies reported the YPLL rate, and eight studies reported YPLL per death. The selected studies that reported the YPLL rate represent the regions of America (Brazil, USA, and Canada), European region (Belgium, Poland, Slovenia, and Switzerland), the Western Pacific region (Australia), Southeast Asia (India), and African region (South Africa). **Fig 3** shows the median YPLL rates by country level, demonstrating that the highest YPLL rate was in Brazil and the lowest was in Switzerland. Meanwhile, for YPLL per death, only three regions are represented: America (Brazil and Canada), Europe (Serbia), and the Western Pacific region (Australia and Japan). The overall median YPLL rate and YPLL per death were 594.2 per 100,000 and 10.9 years, respectively (**Table 3**). The median YPLL rate shows the highest values from South-East Asia (1205 per 100,000), followed by the American region (1163 per 100,000), and the African region (1140 per 100,000). The overall rate was highest in studies conducted after the year 2000 (687.1 versus 453.4 per 100,000). IHD revealed a higher median YPLL rate and YPLL per death compared to cerebrovascular disease in respect to CVD types. Regarding sex, the male showed a higher median YPLL value than the female (**Table 3**).

## Standard expected years of life lost (SEYLL) from CVD

A total of 23 studies reported premature CVD mortality using the SEYLL method from GBD. Eighteen studies calculated the SEYLL rate, and 11 studies calculated SEYLL per death (**Table 4**). The overall median SEYLL rate and SEYLL per death were 1357 per 100,000 and

**Table 2. Characteristics of selected studies.**

| No | 1st author, publication year | Study time | Study design | Follow up | Country | Population | Age range | Data source | CVD type | Method/ Formula | YLL indicators | S | C | O | Total score |
|---|---|---|---|---|---|---|---|---|---|---|---|---|---|---|---|
| | | | | | | | | | | | | **Quality assessment (NOS)** | | | |

**Years of potential life lost (YPLL)**

| No | 1st author, publication year | Study time | Study design | Follow up | Country | Population | Age range | Data source | CVD type | Method/ Formula | YLL indicators | S | C | O | Total score |
|---|---|---|---|---|---|---|---|---|---|---|---|---|---|---|---|
| 1 | Istilli, 2020 [19] | 2010–2014 | cross sectional | N/A | Brazil | General (state of São Paulo) | 30–69 | Vital registration | All CVD | Gardner (1990) | Total YPLL, YPLL rate, YPLL per death | *** | * | *** | 7 / Good |
| 2 | Dolicanin, 2016 [20] | 1992–2013 | cross sectional | N/A | Serbia | General | all | Vital registration | Stroke | Gardner (1990) | Total YPLL, YPLL per death | *** | * | *** | 7 / Good |
| 3 | Krzyżak, 2015 [21] | 2002; 2011 | cross sectional | N/A | Poland | General | 1–69 | Vital registration | All CVD, IHD, Cerebro. | Gardner (1990) | YPLL rate | *** | * | *** | 7 / Good |
| 4 | Dubey, 2014 [22] | 1991–2011 | cross sectional | N/A | India | General | 15–65 | Vital registration | All CVD | Gardner (1990) | Total YPLL, YPLL rate | *** | * | *** | 7 / Good |
| 5 | Góźdź, 2013 [23] | 2002–2010 | cross sectional | N/A | Poland | General (Swietokrzyskie Province) | <70 | Vital registration | All CVD | Gardner (1990) | YPLL rate | *** | * | *** | 7 / Good |
| 6 | Savidan, 2010 [24] | 1995; 2006 | cross sectional | N/A | Switzerland | General | 1–69 | Vital registration | IHD, Cerebro. | Gardner (1990) | YPLL rate | *** | * | *** | 7 / Good |
| 7 | Lam, 2004 [25] | 2001 | cross sectional | N/A | Australia | General | 0–64 | Vital registration | All CVD | Gardner (1990) | Total YPLL, YPLL rate, YPLL per death* | *** | * | *** | 7 / Good |
| 8 | Lessa, 2002 [26] | 1979; 1998 | cross sectional | N/A | Brazil | General | 20–59 | Vital registration | Coronary Heart Disease | Gardner (1990) | Total YPLL, YPLL per death* | *** | * | *** | 7 / Good |
| 9 | Semerl, 2002 [27] | 1998 | cross sectional | N/A | Slovenia | General | <65 | Vital registration | All CVD; IHD, Cerebro. | Gardner (1990) | Total YPLL, YPLL rate | *** | * | *** | 7 / Good |
| 10 | Humblet, 2000 [28] | 1974–1994 | cross sectional | N/A | Belgium | General | 1–64' | Vital registration | IHD | Gardner (1990) | YPLL rate | *** | * | *** | 7 / Good |
| 11 | Yoshida, 1997 [29] | 1984 | cross sectional | N/A | Japan | General (Male, Japanese employees) | 18–64 | Vital registration Ministry of Health & Welfare | HD, Cerebro. | Gardner (1990) | Total YPLL, YPLL per death* | *** | * | *** | 7 / Good |

(*Continued*)

Table 2. (Continued)

| No | 1st author, publication year | Study time | Study design | Follow up | Country | Population | Age range | Data source | CVD type | Method/ Formula | YLL indicators | Quality assessment (NOS) | | | |
|---|---|---|---|---|---|---|---|---|---|---|---|---|---|---|---|
| | | | | | | | | | | | | S | C | O | Total score |
| 12 | Cunningham, 1996 [30] | 1979–1991 | cross sectional | N/A | Australia | General (Aboriginal) | 15–64 | Vital registration | IHD | Gardner (1990) | Total YPLL, YPLL rate | *** | * | *** | 7 — Good |
| 13 | Wigle, 1990 [31] | 1969–1986 | cross sectional | N/A | Canada | General | <75 | Vital registration | ALL CVD | Gardner (1990) | YPLL per death | *** | * | **Unclear formula used | 6 Satisfactory |
| 14 | Mettlin, 1989 [32] | 1970; 1985 | cross sectional | N/A | USA | General | 0–64 | CDC, USA | HD | Gardner (1990) | YPLL rate | ** / N/A data source | * | *** | 6 Satisfactory |
| 15 | Wyndham, 1981 [33] | 1970; 1976 | cross sectional | N/A | South Africa | General | 15–64 | Not reported | All CVD, IHD, Cerebro. | Gardner (1990) | YPLL rate | ** / N/A data soruce | * | *** | 6 Satisfactory |
| 16 | Ouellet, 1979 [34] | 1974 | cross sectional | N/A | Canada | General | 1–69' | Vital registration | IHD, Cerebro. | Gardner (1990) | Total YPLL, YPLL per death* | *** | * | *** | 7 — Good |
| 17 | Romeder, 1977 [9] | 1974 | cross sectional | N/A | Canada | General (Ontario, male) | 1–69' | Not reported | IHD | Gardner (1990) | Total YPLL, YPLL rate, YPLL per death* | *** | * | *** | 7 — Good |
| **Standard expected years of life lost (SEYLL)** | | | | | | | | | | | | | | | |
| 1 | Wang, 2021 [35] | 2005; 2010; 2015 | cross sectional | N/A | China | General | all | Vital registration | All CVD, IHD, Stroke | GBD (1996) | Total SEYLL, SEYLL rate, SEYLL per death* | *** | * | *** | 7 — Good |
| 2 | Wengler, 2021 [36] | 2017 | cross sectional | N/A | Germany | General | all | Vital registration | IHD, Stroke | GBD (1996) | Total SEYLL, SEYLL per death* | *** | * | *** | 7 — Good |
| 3 | Martinez, 2019 [37] | 2000; 2015 | cross sectional | N/A | USA | General | all | GHE, WHO | IHD, Stroke | GBD (1996) | SEYLL rate | *** | * | *** | 7 — Good |
| 4 | Pikala, 2017 [38] | 2013 | cross sectional | N/A | Poland | General | all | Vital registration | All CVD, IHD, Stroke | GBD (1996) | SEYLL rate | *** | * | *** | 7 — Good |
| 5 | Pikala, 2017 [39] | 2000–2014 | cross sectional | N/A | Poland | General | all | Vital registration | All CVD | GBD (1996) | SEYLL rate | *** | * | *** | 7 — Good |

(Continued)

**Table 2.** (Continued)

| No | 1st author, publication year | Study time | Study design | Follow up | Country | Population | Age range | Data source | CVD type | Method/ Formula | YLL indicators | S | C | O | Total score |
|---|---|---|---|---|---|---|---|---|---|---|---|---|---|---|---|
| | | | | | | | | | | | | Quality assessment (NOS) | | | |
| 6 | Takslerx, 2017 [40] | 1995; 2015 | cross sectional | N/A | USA | General | all | Vital registration | Heart Disease, Cerebro. | GBD (1996) | Total SEYLL, SEYLL per death* | *** | * | *** | 7 / Good |
| 7 | Bryla, 2016 [41] | 1999–2011 | cross sectional | N/A | Poland | General | all | Vital registration | All CVD, IHD, Cerebro. | GBD (1996) | SEYLL rate; SEYLL per death | *** | * | *** | 7 / Good |
| 8 | Lee, 2016 [42] | 2012 | cross sectional | N/A | Korea | General | all | Vital registration | IHD | GBD (1996) | SEYLL rate | *** | ** Adj. DAW | *** | 8 / Good |
| 9 | Maniecka-Bryla, 2015 [43] | 2011 | cross sectional | N/A | Poland | General | all | Vital registration | All CVD, IHD, Cerebro. | GBD (1996) | Total SEYLL, SEYLL rate, SEYLL per death | *** | * | *** | 7 / Good |
| 10 | Cheng, 2013 [44] | 2008–2010 | cross sectional | N/A | China | General (Hube) | >15 | DSPs system | All CVD, IHD, Cerebro. | GBD (1996) | Total SEYLL, SEYLL rate, SEYLL per death | *** | * | *** | 7 / Good |
| 11 | Maniecka-Bryla, 2012 [45] | 2008 | cross sectional | N/A | Poland | General (Łódź province) | all | Vital registration | All CVD | GBD (1996) | Total SEYLL, SEYLL rate | *** | * | *** | 7 / Good |
| 12 | Gènova-Maleras, 2011 [46] | 2008 | cross sectional | N/A | Spain | General | all | Vital registration | All CVD, IHD, Cerebro. | GBD (1996) | SEYLL rate | *** | ** Adj. DAW | *** | 8 / Good |
| 13 | Vijitsoonthronkul, 2011 [47] | 1997; 2006 | cross sectional | N/A | Thailand | General | all | Vital registration | All CVD, IHD, Cerebro. | GBD (1996) | SEYLL rate | *** | * | *** | 7 / Good |
| 14 | Plass, 2013 [48] | 2010 | cross sectional | N/A | Hong Kong | General | all | Vital registration | ALL CVD | GBD (1996) | Total SEYLL, SEYLL rate, | *** | ** Adj. DAW | *** | 8 / Good |
| 15 | Milicevic, 2009 [49] | 2000 | cross sectional | N/A | Serbia | General | all | Vital registration | All CVD | GBD (1996) | Total SEYLL, SEYLL rate, SEYLL per death* | *** | ** Adj. DAW | *** | 8 / Good |

(Continued)

Table 2. (Continued)

| No | 1st author, publication year | Study time | Study design | Follow up | Country | Population | Age range | Data source | CVD type | Method/ Formula | YLL indicators | Quality assessment (NOS) | | | |
|---|---|---|---|---|---|---|---|---|---|---|---|---|---|---|---|
| | | | | | | | | | | | | S | C | O | Total score |
| 16 | Aragon, 2008 [50] | 2003–2004 | cross sectional | N/A | USA | General (San Francisco) | all | Vital registration | IHD, Cerebro. | GBD (1996) | Total SEYLL, SEYLL rate, SEYLL per death | *** | ** Adj. DAW | *** | 8 Good |
| 17 | Cai, 2008 [51] | 1998–2003 | cross sectional | N/A | China | General (Gan Du) | all | CDC, Guan Du | IHD, Stroke | GBD (1996) | SEYLL rate | *** | **Adj. DAW | *** | 8 Good |
| 18 | Vlajinac, 2008 [52] | 2000 | cross sectional | N/A | Serbia | General | all | Vital registration | All CVD | GBD (1996) | Total SEYLL, SEYLL rate | *** | **Adj. DAW | *** | 8 Good |
| 19 | Cai, 2006 [53] | 2003 | cross sectional | N/A | China | General (Kunmin, Yunnan) | all | District CDC & Shin Lin Hospital | IHD, Stroke | GBD (1996) | SEYLL rate | *** | **Adj. DAW | *** | 8 Good |
| 20 | Lapostolle, 2008 [54] | 2000–2002 | cross sectional | N/A | France | General | all | CepiDC, INSERM | All CVD | GBD (1996) | Total SEYLL, SEYLL per death* | *** | * | *** | 7 Good |
| 21 | Marshall, 2004 [55] | 1990–1996 | cross sectional | N/A | New Zealand | General (Hunan) | all | Vital registration | IHD, Stroke | GBD (1996) | SEYLL per death | *** | **Adj. DAW | *** | 8 Good |
| 22 | Mariotti, 2003 [56] | 1998 | cross sectional | N/A | Italy | General | all | Vital registration | IHD, Stroke | GBD (1996) | Total SEYLL, SEYLL per death* | *** | **Adj. DAW | *** | 8 Good |
| 23 | Indrayan, 2002 [57] | 1995 | cross sectional | N/A | India | General (rural) | all | Survey SCD | HD | GBD (1996) | SEYLL rate | *** | * | *** | 7 Good |

* NOS: Newcastle-Ottawa Scale (S: selection, C: comparability, and O: outcome); Adj. DAW: Adjusted Discounting age weight (SEYLL was adjusted with time discounting and/or age-weighting); Vital registration, including the national death registry, mortality database, or censuses; GBD: Global Burden of Disease Study; GHE, WHO: Global Health Estimates, World Health Organization; DSPS: Disease Surveillance Points System; CDC: Center for Disease Control; CepiDC, INSERM: Center for Epidemiology of the Medical Causes of Death (CepiDc), a department of the National Institute on Health and Medical Research (INSERM); Survey SCD: Survey of Causes of Death; CVD: cardiovascular disease; Cerebro: cerebrovascular disease; SEYLL: standard expected years of life lost; YPLL: years of potential life lost; SEYLL per death* or YPLL per death* were estimated values manually calculated from total YLL divided by the number of CVD deaths, whenever data is available.

**Table 3. Summary of total years of potential life lost (YPLL) from CVD mortality according to study characteristics.**

| Characteristics | | Total | | Data time | |
|---|---|---|---|---|---|
| | | | | Year 2000–2022 | Year 1970–1999 |
| | n | Median (IQR) | | Median (IQR) | Median (IQR) |
| **YPLL rate per 100,000** | 12 | 594.2 (163.4, 992.8) | | 687.1 (271.3, 1091.8) | 453.4 (135.5, 845.0) |
| WHO regions | | | | | |
| African | 1 | 1,140.0 (992.5, 2,228.8) | | N/A | 1,140.0 (992.5, 2,228.8) |
| America | 3 | 1,163.1 (1,105.6, 1,590.0) | | 1,733.5 (1,419.6, 2,047.5) | 1,163.1 (979.5, 1,376.6) |
| Europe | 5 | 340.6 (118.6, 683.0) | | 556.9 (180.9, 963.6) | 166.5 (100.3, 453.4) |
| South-East Asia | 1 | 1,205.0 (1,077.5, 1,332.5) | | 1,205.0 (1,077.5, 1,332.5) | N/A |
| Western Pacific | 2 | 900.0 (766.7, 1,078.0) | | 766.7 (612.0, 922.4) | 2,200.0 (1,550.0, 2,850.0) |
| CVD types | | | | | |
| All CVD | 7 | 1,078.0 (687.1, 2,282.4) | | 1,078.0 (702.8, 2,069.1) | 2,000.2 (1,341.6, 2,658.9) |
| Cerebrovascular disease | 4 | 154.2 (84.2, 293.6) | | 200.1 (89.7, 317.0) | 135.5 (100.3, 246.6) |
| IHD | 8 | 453.0 (131.4, 822.0) | | 243.2 (136.4, 512.9) | 476.2 (143.4, 874.0) |
| Sex | | | | | |
| Female | 9 | 160.3 (94.4, 607.5) | | 457.4 (154.2, 687.1) | 103.8 (89.2, 135.7) |
| Male | 10 | 753.8 (398.2, 1,722.9) | | 1,078.0 (585.0, 2,282.4) | 476.2 (371.0, 716.2) |
| **YPLL per death** | 8 | 10.9 (4.7, 11.1) | | 4.3 (3.0, 11.2) | 10.9 (10.6, 11.2) |
| WHO regions | | | | | |
| African | 0 | N/A | | N/A | N/A |
| America | 5 | 11.0 (10.8, 11.2) | | 11.7 (11.4, 12.0) | 10.9 (10.7, 11.1) |
| Europe | 1 | 3.2 (3.0, 3.6) | | 3.0 (3.0, 3.2) | 4.0 (3.8, 4.3) |
| South-East Asia | 0 | N/A | | N/A | N/A |
| Western Pacific | 2 | 16.3 (10.7, 17.1) | | 5.0 (5.0, 5.0) | 17.1 (16.7, 17.5) |
| CVD types | | | | | |
| All CVD | 6 | 11.0 (11.0, 11.7) | | 11.4 (9.5, 11.9) | 11.0 (11.0, 11.0) |
| Cerebrovascular disease | 4 | 3.5 (3.0, 6.2) | | 3.0 (3.0, 3.2) | 7.9 (4.3, 12.4) |
| IHD | 7 | 10.9 (10.6, 11.1) | | N/A | 10.9 (10.6, 11.1) |
| Sex | | | | | |
| Female | 8 | 10.6 (4.1, 10.9) | | 3.6 (3.3, 7.3) | 10.8 (9.1, 11.0) |
| Male | 8 | 10.9 (5.3, 11.9) | | 3.0 (3.0, 7.6) | 11.0 (10.7, 13.7) |

*n, number of studies reported YPLL values. N/A, not applicable.

Each study reported either the YPLL rate or the YPLL per death or both. They may provide YPLL values for each CVD type and sex or for the overall population. We treat each value as separate data for each paper to calculate the median and IQR (Interquartile range).

13.8 years, respectively, where both values were higher in studies conducted after the year 2000. Most of the studies that reported the SEYLL rate were from the European region (Poland, Serbia, and Spain) and the Western Pacific region (China, Hong Kong, South Korea, and Thailand). Only one study came from Southeast Asia (India) and two from the Americas (both from the United States). There were no studies reported on SEYLL in Africa or the Eastern Mediterranean Region. The SEYLL rates for each country were presented in **Fig 4**. South Korea had the lowest SEYLL rate, while Serbia had the highest SEYLL rate, followed by Hong Kong, China, and Thailand. In terms of sex, males demonstrated more SEYLL as compared with females for all studies conducted before and after the year 2000 (**Table 4**). For CVD types, IHD has a lower median SEYLL rate than cerebrovascular disease for all studies conducted prior to and following 2000 (**Table 4**).

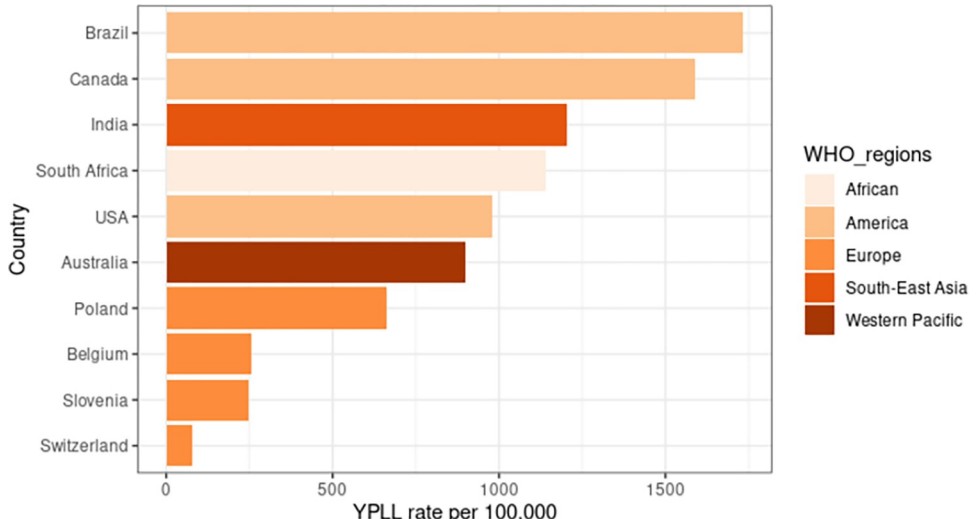

**Fig 3. The YPLL rate of countries with available data.**

## The pattern of years of life lost from CVD

To analyse this pattern further, we stratified the YPLL rate and SEYLL rate according to CVD type (**Fig 5**) and sex (**Fig 6**) and visualised them with different colours for each WHO region. To be more comparable with SEYLL, we selected the studies for YPLL from the past three decades (1990–2022), since no study for SEYLL was reported before 1990. Our review discovered that the pattern of YPLL and SEYLL rates from total CVD has increased slightly during the past three decades (**Fig 5**). However, after stratification with specific CVD types, the individual rate of IHD and cerebrovascular disease was nearly flat and did not demonstrate a substantial increase over time. Meanwhile, the pattern of the SEYLL rate for cerebrovascular disease has declined slightly (**Fig 5B**). However, this result should be interpreted with caution considering where the only studies reporting the specific CVD type for the YPLL rate were from Europe (displayed as a green dot in **Fig 5A**), but for the SEYLL rate, data was originated from the Western Pacific, Europe, and other regions (**Fig 5B**). In terms of sex, males demonstrated a substantially increased rate for the past three decades, especially for the YPLL rate. However, for females, both the YPLL and SEYLL rates demonstrated only marginal increases after three decades (**Fig 6A** and **6B**). Due to the lack of data, we were only able to stratify the analysis by geographic region using the SEYLL method. Interestingly, although the overall SEYLL rate has increased slightly over the past three decades, the European region has experienced a decline, and the American region has nearly reached a plateau. While the rate in the Western Pacific region demonstrated a notable increase (**Fig 7**).

## Discussion

To our knowledge, this is the first systematic review examining CVD mortality using the difference method of YLL as an indicator for premature mortality. Although GBD is the most widespread global estimate of the burden of disease and provides a comprehensive picture of the health of populations, the GBD study relies on various mathematical models and assumptions with absent or limited quality mortality data in some countries, particularly in low-income countries [58, 59]. Our review, however, extends the existing literature by reporting the year of life lost methods for premature CVD mortality using observed data (mostly registry

**Table 4. Summary of total standard expected years of life lost (SEYLL), from CVD mortality method from GBD study (1996) according to study characteristics.**

| Characteristics | | SEYLL rate per 100,000 | | |
| --- | --- | --- | --- | --- |
| | | Total | Data time | |
| | | | 2000–2022 | 1970–1999 |
| | n | Median (IQR) | Median (IQR) | Median (IQR) |
| SEYLL rate per 100,000 | | | | |
| Total | 18 | 1357.0 (723.7, 2518.4) | 1490.0 (728.2, 2635.7) | 904.0 (512.2, 1314.2) |
| WHO regions | | | | |
| America | 2 | 925.0 (594.5, 1,364.6) | 925.0 (594.5, 1,364.6) | N/A |
| Europe | 8 | 1,546.5 (817.5, 4,834.0) | 1,546.5 (817.5, 4,834.0) | N/A |
| South-East Asia | 1 | 850.0 (850.0, 850.0) | N/A | 850.0 (850.0, 850.0) |
| Western Pacific | 7 | 1,363.5 (726.6, 1,996.1) | 1,540.0 (830.0, 2,426.8) | 958.0 (470.0, 1,351.0) |
| CVD types | | | | |
| All CVD | 12 | 4,337.0 (1,903.1, 5,893.0) | 4,516.2 (2,208.7, 5,917.1) | 958.0 (958.0, 958.0) |
| Cerebrovascular disease | 10 | 1,165.1 (750.3, 1,462.5) | 1,083.0 (454.8, 1,458.4) | 1,360.5 (1,240.8, 1,445.0) |
| IHD | 13 | 731.5 (466.7, 1,635.3) | 843.5 (550.0, 1,723.6) | 470.0 (446.5, 726.5) |
| Sex | | | | |
| Female | 16 | 1,170.2 (615.0, 2,045.0) | 1,284.2 (638.9, 2,841.8) | 759.0 (444.8, 1,188.8) |
| Male | 16 | 1,558.2 (805.5, 2,933.6) | 1,723.7 (1,015.5, 4,146.0) | 1,082.5 (728.0, 1,355.8) |
| SEYLL per death | | | | |
| Total | 11 | 13.8 (9.7, 19.1) | 15.4 (9.7, 19.4) | 11.5 (9.8, 13.8) |
| WHO regions | | | | |
| America | 2 | 9.4 (7.9, 10.7) | 8.6 (7.2, 10.2) | 10.4 (10.2, 10.6) |
| Europe | 6 | 8.8 (5.9, 9.7) | 8.8 (5.8, 9.9) | 8.9 (8.7, 9.1) |
| South-East Asia | 0 | N/A | N/A | N/A |
| Western Pacific | 3 | 18.4 (14.7, 20.3) | 18.6 (17.0, 20.7) | 13.8 (13.3, 14.1) |
| CVD types | | | | |
| All CVD | 4 | 16.9 (9.0, 19.5) | 16.9 (9.0, 19.5) | N/A |
| Cerebrovascular disease | 9 | 12.2 (9.8, 18.9) | 13.9 (10.2, 19.7) | 11.1 (9.6, 12.6) |
| IHD | 9 | 13.8 (9.9, 18.2) | 14.4 (9.9, 19.0) | 12.3 (10.5, 13.9) |
| Sex | | | | |
| Female | 8 | 12.6 (7.9, 17.0) | 11.5 (7.2, 17.1) | 13.8 (13.7, 13.9) |
| Male | 8 | 14.2 (9.8, 21.3) | 14.2 (9.0, 21.5) | 13.4 (12.8, 14.1) |

* n, number of studies reported SEYLL values. N/A, not applicable

Each study reported either the SEYLL rate or the SEYLL per death or both. They may provide SEYLL values for each CVD type and sex or for the overall population. We treat each value as separate data for each paper to calculate the median and IQR (interquartile range).

data) from selected primary studies. In general, we synthesised all studies that reported premature CVD mortality based on two commonly used YLL formulas, the YPLL and SEYLL methods. Despite SEYLL is the most updated formula, we included YPLL in our synthesis since numerous studies still used the YPLL method after 1990. Interestingly, our review found that the YPLL and SEYLL methods produce almost similar results, particularly in terms of the YLL rate pattern. Over the last three decades, we found that the pattern of YPLL and SEYLL rates for overall CVD has increased slightly (1990–2021). However, the stratification analysis for specific CVD types does not demonstrate a substantial increase over time. Even though this review could not measure the magnitude of the change in trend, our descriptive findings identified a slight reduction in the SEYLL rate for cerebrovascular disease (the main CVD types)

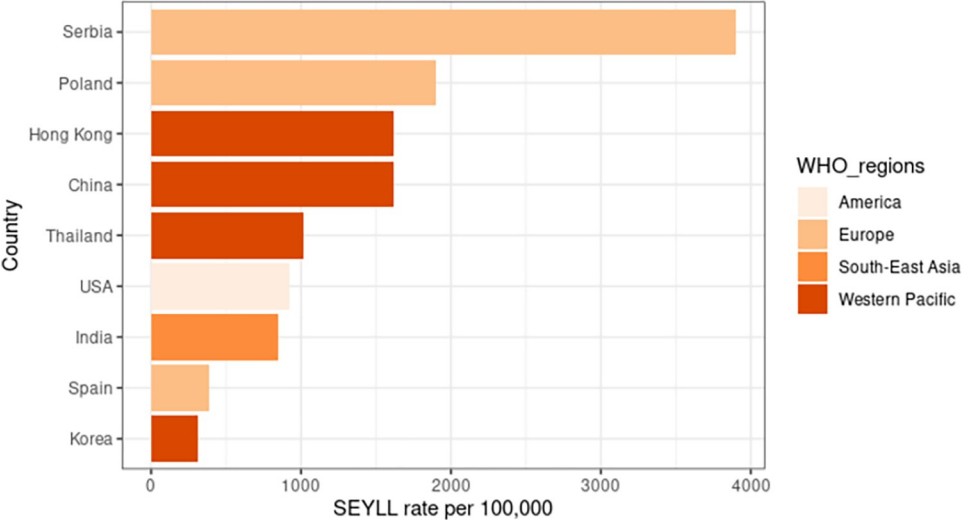

**Fig 4. The SEYLL rate of countries with available data.**

(a) Pattern of years of potential life lost (YPLL) per 100,000 from 1990 to 2022 by CVD types

(b) Pattern of standard expected years of life lost (SEYLL) per 100,000 from 1990 to 2022 by CVD types

**Fig 5.** Patterns of YPLL rate (a) and SEYLL rate (b) according to CVD types and WHO regions.

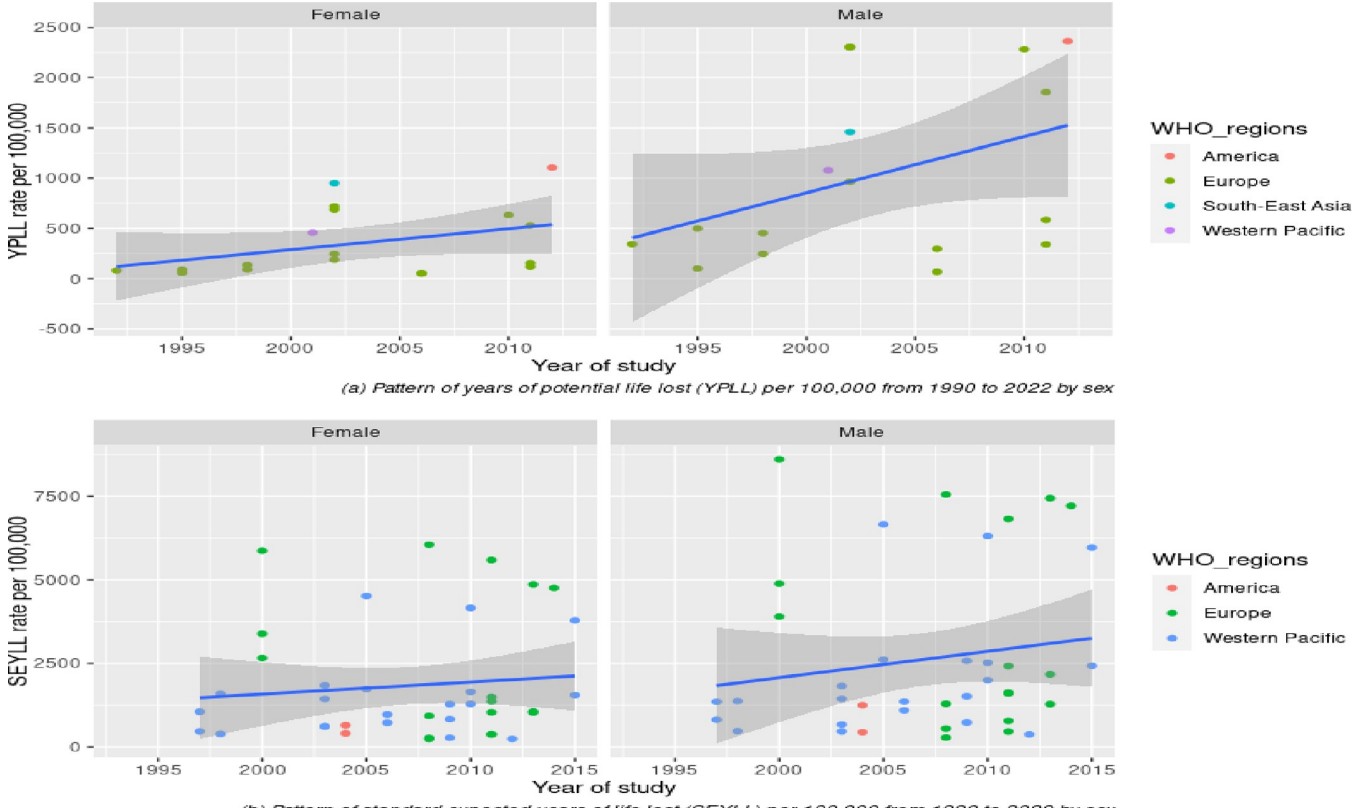

**Fig 6.** (a): Patterns of YPLL rate (a) and SEYLL rate (b) according to sex and WHO regions.

after three decades. This finding is in line with the reported results from a comprehensive analysis of the GBD study 2019, which show the YLL trend for CVD has been decreasing globally over the last three decades [60].

Throughout the synthesis, we also discovered that the pattern of the YLL rate increased substantially among men compared to women. For all studies conducted before and after 2000, the YPLL rate and SEYLL rate from CVD in men were also higher than in women. The results

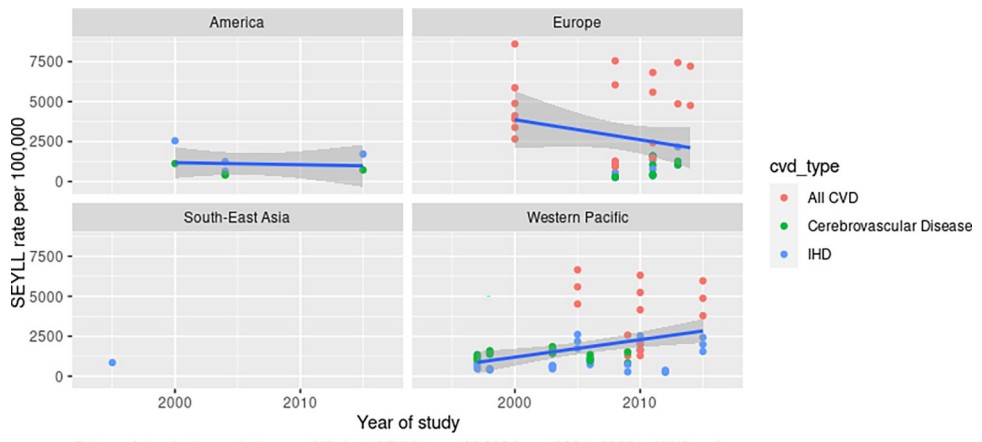

**Fig 7. Patterns of SEYLL rate by WHO regions.**

of our review were in accordance with a previous study using GBD estimated data, showing men had higher values of CVD YLL than women [60–62]. A previous study using Global Health Estimates (GHE) also reported that the overall premature deaths from CVD were 35.6% higher in men than women [63]. In recent decades, several studies have discussed the sex disparities related to premature CVD mortality, including sex differences in CVD presentation, treatment, and outcomes [64–66]. A clear understanding of sex disparities in premature CVD mortality, specifically at the country level, is essential in order to develop detailed and effective CVD prevention and control strategies.

Geographically, we found the Southeast Asia region, which is represented by low- and middle-income countries (LMICs), demonstrated the highest YPLL rate from total CVD. Consistently with the study using global data from GHE, they demonstrated that Southeast Asia was the second highest premature CVD death rate after the Eastern Mediterranean [63]. Our analysis also showed other LMICs, including Brazil, India, South Africa, and Serbia, were among those with the highest YLL rates, while the rate was low in high-income countries such as Switzerland, Belgium, Spain, Slovenia, the USA, and South Korea. Our findings support the WHO report that LMICs account for at least three-quarters of CVD-related premature mortality [67, 68]. The other systematic review on cost-effectiveness also found that the substantial burden of CVD remains higher in LMICs than in high-income countries [69].

A systematic analysis using GBD data by Roth et al. demonstrated the largest increase in premature mortality attributable to CVD in the past two decades was in South, East, and Southeast Asia, and parts of Latin America [1]. Although global CVD mortality rates have declined dramatically, they reveal that the number of life years lost to premature CVD deaths is increasing in LMICs. Due to data limitations, our review could only summarise the SEYLL pattern in Europe, the Western Pacific, and the Americas. Interestingly, this review was almost consistent with the study by Roth et al. [1] where our synthesis found the European countries (mostly represented by high-income countries) have shown a declining pattern while, the rate in the Western Pacific region (mostly from LMICs) showed an increase over the past three decades. Previous studies reported that premature mortality tends to be higher in low-resource areas, especially in low-income countries, due to the management and treatment of the preclinical population and the diagnosed CVD patients are less advanced in low-resource areas than in developed areas [70, 71]. The changing pattern of NCD risk factors in developing countries, in the Western Pacific region could be attributed to the increasing burden of NCDs, in particular CVD mortality [72]. On the other hand, decades of effort in NCD risk prevention have resulted in a progressive decline in premature mortality. Therefore, LMICs shall dwell on and scale-up six cost-effective intersectoral policies to intervene with the behavioural risks (e.g., tobacco smoking, harmful use of alcohol, and excess sodium intake), starting in 2023, and sustain them through 2030 to reduce premature mortality from CVD over the next decade [73]

Policies and health interventions need to be scaled and adjusted for a wide range of local conditions to achieve the health goals set by the United Nations. Countries and health care systems need to concentrate on delivering efficient interventions to reverse these trends, especially in the post COVID-19 pandemic. The COVID-19 pandemic has been shown to have a significant impact on premature mortality [74–76]. Reducing premature CVD mortality during the COVID-19 pandemic is a critical challenge that demands a comprehensive, multidisciplinary approach. Further research is necessary to emphasize the significance of addressing the impact of the pandemic on CVD in order to reduce premature mortality. Continuous monitoring of years of life lost using either the YPLL or SEYLL methods is important to gain a deeper understanding of the burden of premature CVD mortality and to guide efforts to prevent and treat CVD in the post-pandemic era.

Several important limitations to this review should be noted, which may limit the applicability of the results. First, publication bias is always an issue in systematic reviews. We endeavoured to address this by obtaining data from all available sources, including those from electronic databases, citations, and authors. Second, there is missing information from some (state how many) countries, thereby limiting global representativeness. However, most studies in this review used the registry data that represents the country's burden. In addition, we considered the correct data source to be included in this review by excluding studies that used estimated data and studies with non-nationally representative populations. Third, some studies in our review are from different time periods. As a result, we were unable to identify a global study conducted during the same time period. Fourth, as with other review synthesis studies, it is always impossible to obtain complete data that are relevant to the objective of the analysis. Our review faced limited data points for assessing the trend of premature CVD mortality with different sexes and CVD types in each region. In order to address this issue, we evaluated the trend by stratifying the study time by region and sex. The results of the trend were also analysed from the observed data. We did not include data from GBD studies as their results were estimated (or predicted). Fifth, the different designs and characteristics, including age coverage, study time, and methods of calculation for YLL, of all the included studies may lead to high heterogeneity, which, in turn, may lower the quality of the evidence in this review. We performed the subgroup analysis of synthesis according to the YLL method, study time, regional area, CVD types, and sex to reduce the effect of heterogeneity between studies.

## Conclusion

This systematic review provides an overview of premature CVD mortality for monitoring purposes, tracking progress, and advocating for resources and policy attention. In summary, the results of the current review indicate that the United Nations Sustainable Development Goal to reduce premature mortality due to CVD by 25% by 2025 will be challenging, especially for countries of low and middle income. The increasing burden among men, and in LMICs points glaringly towards the need for more cost-effective treatment and prevention strategies. A global focus should be directed at reversing these trends, including those that control and prevent diabetes, reduce obesity and high cholesterol, improve diet and exercise, and reduce excessive alcohol and tobacco use.

## Supporting information

**S1 Checklist.**
(DOCX)

**S2 Checklist. PRISMA checklist.**
(DOCX)

**S1 Table. Search terms.**
(DOCX)

**S2 Table. Data extraction form.**
(XLSX)

**S3 Table. NOS assessment.**
(DOCX)

## Acknowledgments

We would like to thank the Director-General of Health Malaysia for his permission to publish this article.

## Author Contributions

**Conceptualization:** Wan Shakira Rodzlan Hasani, Nor Asiah Muhamad, Tengku Muhammad Hanis, Nur Hasnah Maamor, Chen Xin Wee, Mohd Azahadi Omar, Shubash Shander Ganapathy, Zulkarnain Abdul Karim, Kamarul Imran Musa.

**Formal analysis:** Wan Shakira Rodzlan Hasani.

**Investigation:** Wan Shakira Rodzlan Hasani, Tengku Muhammad Hanis, Chen Xin Wee.

**Methodology:** Wan Shakira Rodzlan Hasani, Nor Asiah Muhamad, Tengku Muhammad Hanis, Mohd Azahadi Omar, Kamarul Imran Musa.

**Project administration:** Wan Shakira Rodzlan Hasani, Nor Asiah Muhamad, Nur Hasnah Maamor.

**Resources:** Wan Shakira Rodzlan Hasani, Nor Asiah Muhamad.

**Supervision:** Nor Asiah Muhamad, Tengku Muhammad Hanis, Kamarul Imran Musa.

**Validation:** Tengku Muhammad Hanis, Mohd Azahadi Omar, Kamarul Imran Musa.

**Visualization:** Wan Shakira Rodzlan Hasani, Tengku Muhammad Hanis.

**Writing – original draft:** Wan Shakira Rodzlan Hasani.

**Writing – review & editing:** Wan Shakira Rodzlan Hasani, Nor Asiah Muhamad, Tengku Muhammad Hanis, Nur Hasnah Maamor, Chen Xin Wee, Mohd Azahadi Omar, Shubash Shander Ganapathy, Zulkarnain Abdul Karim, Kamarul Imran Musa.

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
