## [Decision Letter · Decision Letter 0]

6 Jan 2023

PONE-D-22-34213The burden of premature mortality from cardiovascular diseases: a systematic review of years of life lostPLOS ONE

Dear Dr. Hasani,

Thank you for submitting your manuscript to PLOS ONE. After careful consideration, we feel that it has merit but does not fully meet PLOS ONE’s publication criteria as it currently stands. Therefore, we invite you to submit a revised version of the manuscript that addresses the points raised during the review process.

According to the precious comments of the reviewers, the submitted manuscript has investigated an interesting notion; however, several methodological issues need be considered to improve the conducted study. I would like to ask you to follow each comment carefully and address them to improve the provided material and help to reaching a final decision on this submission.

We look forward to receiving your revised manuscript.

Kind regards,

Sina Azadnajafabad, MD, MPH

Academic Editor

PLOS ONE

3. We note that Figures 2, 5a and 5b in your submission contain [map/satellite] images which may be copyrighted. All PLOS content is published under the Creative Commons Attribution License (CC BY 4.0), which means that the manuscript, images, and Supporting Information files will be freely available online, and any third party is permitted to access, download, copy, distribute, and use these materials in any way, even commercially, with proper attribution. For these reasons, we cannot publish previously copyrighted maps or satellite images created using proprietary data, such as Google software (Google Maps, Street View, and Earth). For more information, see our copyright guidelines: http://journals.plos.org/plosone/s/licenses-and-copyright.

a. You may seek permission from the original copyright holder of Figures 2, 5a and 5b to publish the content specifically under the CC BY 4.0 license. 

Reviewers' comments:

Reviewer's Responses to Questions

**Comments to the Author**

1. Is the manuscript technically sound, and do the data support the conclusions?

Reviewer #1: Partly

Reviewer #2: Partly

Reviewer #3: Yes

Reviewer #4: Partly

2. Has the statistical analysis been performed appropriately and rigorously? 

Reviewer #1: N/A

Reviewer #2: Yes

Reviewer #3: N/A

Reviewer #4: N/A

3. Have the authors made all data underlying the findings in their manuscript fully available?

Reviewer #1: Yes

Reviewer #2: Yes

Reviewer #3: Yes

Reviewer #4: Yes

4. Is the manuscript presented in an intelligible fashion and written in standard English?

Reviewer #1: Yes

Reviewer #2: Yes

Reviewer #3: Yes

Reviewer #4: Yes

5. Review Comments to the Author

Reviewer #1: In this study, Hasani. presented a systematic review on the measures of the premature mortality of cardiovascular disease (CVD) worldwide. They highlighted differences between different methods for calculation of YLL which was interesting. They found that low- and middle-income countries (LMIC) are more being affected by CVD premature mortality compared with high-income countries (HIC). I have the following comments:

1) The authors should distinguish their findings in the context of the current literature. The main finding (higher premature CVD mortality in LMIC compared with HIC) has been known more or less in the last decades. Therefore, the authors should highlight the specific contribution of their study. In my opinion, the distinction of YPLL and SEYLL may be an added value.

2) Why did the authors include non-nationally representative studies in their evidence synthesis? Studies without a specified data source, or those with a cohort design may have biased results. The premature mortality of a region should not be inferred from such studies. Furthermore, the findings from Fene 2020 that used GBD 2017 cannot be compared with other studies that are based on vital registration data. Data source of the studies should be strongly considered for correct evidence synthesis.

3) There are not enough data points for assessing the trend of premature mortality in each region or in the world. If the authors utilized findings from the GBD study, it would make sense to present the trend.

4) The first paragraph of the results section has redundant data with Figure 1 and should be summarized.

5) The conclusion paragraph should be presented as the 'Implication' paragraph in the discussion section. The conclusion section should be the gist of the manuscript, with limited discussion of the findings or their implications.

6) In contrast with 'Gender', 'Sex' is biologically determined, and it is recommended to use 'Sex' instead of 'Gender' in scientific texts.

Reviewer #2: Dear Editor,

Many thanks for this precious opportunity you gave me to review this paper.

The authors made a systematic review on literatures to evaluate the years of potential life lost (YPLL) and standard expected years of life lost (SEYLL) of premature cardiovascular disease.

The study is well conducted and results presentation is satisfactory, however, it should be interpreted in the context of several possible limitations which could be overcome with a major revision:

1- Authors should state the “Premature CVD death” cut-off. It usually considers as 45/55 in females/males, but it varies from 45-65 in literatures. Although the authors stated the definition of premature mortality in different studies on “Results, Line 40”, but the main definition in the current study should be clear.

2- Authors assessed the pattern of years of life lost from CVD, which I believe that it includes a thoughtful error in the term of generalization. In each decades the available studies are different. This trend could not be explained by the total number of events, it would be due to the studies region in each decade. To be clearer, YPLL in 1990-2000, measured from one study, which conducted in Europe. These results could not be declared as the “trend of premature CVDs”.

3- To evaluate the trend, I suggest to add a table which have “decades” as the “columns” with “sex” and “WHO Regions” as the “rows”.

4- In the discussion, the key risk factors in LMICs should be noted. Also, mentioning the previous beneficial policies to reduce and prevent the premature CVD deaths in other countries should be examined.

Minor:

1- Introduction: to evaluate the trend of NCDs it would be better to use the most recent GBD studies. Thus, I suggest to replace the reference 2 with the GBD 2019 study.

2- In Introduction, line 133, SEYL should change to SEYLL.

3- I believe the third paragraph of introduction is more about the method and Statistical analysis and should not be presented as introduction.

4- Authors claimed that they aimed to determine global temporal trends in premature CVD mortality, I’m afraid this paper could not reach this goal.

Reviewer #3: In this manuscript, Hasani and colleagues conducted a systematic review to assess years of life lost due to cardiovascular disease by analyzing studies reporting cardiovascular disease-related premature mortality.

This is, as far as I am aware, the first systematic review of this topic (except for the GBD study).

The manuscript, however, cannot be accepted in its current form and requires major revision.

Major Comments:

1- The search strategy might not be inclusive and requires some revision. I suggest that the authors also include non-mesh terms with OR in the first group of keywords in their PubMed search. Some examples of the keywords that can be added are: "Cardiovascular Disease"[Title/Abstract] OR “Heart Disease”[Title/Abstract] OR Aneurysm[Title/Abstract] OR Cardiomyopath*[Title/Abstract] etc. The authors can retrieve and add all of the relevant keywords from the MESH tree. Moreover, I cannot understand why the authors did not choose to search title/abstract for their second group of keywords in PubMed instead of just searching in the titles. I also suggest that the authors avoid using filters on PubMed as it may lead to missing some relevant studies.

2- The authors stated, "We did not include reviews, meta-analyses, letters, comments, or editorials. "Nevertheless, these study types may also contain pertinent data (please see https://doi.org/10.1161/circ.132.suppl 3.17368 for more information on heart failure). I propose that the authors reconsider their decision.

3- In the Prospero protocol, the authors indicated that they would also search Embase and CINAHL. Please include these databases or indicate that you were unable to adhere to the primary protocol.

4- In the results section, the authors have stated in multiple sentences whether the trend has increased or decreased. This study was unable to demonstrate statistical significance for any of these trends.

5- In the discussion, please consider including a section on the potential impact of the COVID-19 pandemic on the burden of premature mortality attributable to cardiovascular diseases. Please investigate whether any studies exist that compare the burden prior to and during the pandemic.

6- The discussion requires further elaboration on preventive measures and efforts that can improve the quality of care in high-income and low-income countries.

Minor comments:

1- Could the authors include the extraction sheet as supplemental material to increase the study's reproducibility?

2- Since few studies were conducted to assess the burden of premature mortality due to HF, I questioned whether the authors attempted to obtain the necessary information from the following paper. http://dx.doi.org/10.1136/heartjnl-2020-317833

3- Please do not use the abbreviated form of "it's" on page 18, line 166.

4- Please note, on page 19, line 197, that according to the GBD study, the rate of All ages global YLL due to cardiovascular diseases did not differ significantly between 2019 and 1990 (95% UI= -0.12 to 0.01 (GBD Results tool))

5-In figure 4, why did the authors choose these countries as opposed to all? Please add the number of available studies from each country to the respective bar.

Reviewer #4: This is a well conducted and well reported systematic review focusing on YLLs of CVD to determine the burden of premature CVD mortality on a global scale.

I have some comments and questions:

- The main issue is to determine how representative these data are of the population in that region/time? I suppose most of the included publications are not done with nationally representative data. Even if they are national, it would be a big assumption to take these estimates to represent a whole WHO region. I think the paper should put less emphasis on the regional results.

- The GBD study provides a great source, which many researchers use. The authors can discuss the advantages and disadvantages of their study, which I understand uses real-world data without modeling, to the GBD estimates, considering that GBD study provides a much more comprehensive picture of the CVD YLLs.

- Median and range are reported for YLLs, yet it is not clear how these were calculated and reported. I am not sure how reporting median/range of included studies’ results, especially in groups with 3 studies or even 1 study, is useful. Moreover, the range for the total number of studies is so wide, it does not convey a clear message. Perhaps the authors can clarify their methods and discuss these limitations.

- Figure 5 shows a world map of SEYLL rates. I think such a figure has a limitation in that these estimates are not from the same time period, but rather come from studies over five decades.

- One of the included studies, from the Caribbean countries, has GBD study 2017 as its source data. Since GBD studies only provide estimates, I think it would be better to exclude this record. Or perhaps there was some reason behind including this study that the authors can clarify.

Minor comments:

- In figure 1, among the reasons for exclusion of full-texts, it seems reporting only absolute YLL is written twice. Please recheck it.

- I think the introduction can be more concise. This would leave room for a more complete discussion of the findings, and what the study adds to the literature.

6. PLOS authors have the option to publish the peer review history of their article (what does this mean?). If published, this will include your full peer review and any attached files.

Reviewer #1: No

Reviewer #2: **Yes: **Ali Sheikhy

Reviewer #3: No

Reviewer #4: No

---

## [Author Response · Author response to Decision Letter 0]

6 Feb 2023

Response to reviewer’s 1 comments

In this study, Hasani. presented a systematic review on the measures of the premature mortality of cardiovascular disease (CVD) worldwide. They highlighted differences between different methods for calculation of YLL which was interesting. They found that low- and middle-income countries (LMIC) are more being affected by CVD premature mortality compared with high-income countries (HIC). I have the following comments:

Author’s response

Dear reviewer, we are grateful for the comment. In response to this comment, we have revamped this review as follows:

No Comment Response

1 The authors should distinguish their findings in the context of the current literature. The main finding (higher premature CVD mortality in LMIC compared with HIC) has been known more or less in the last decades. Therefore, the authors should highlight the specific contribution of their study. In my opinion, the distinction of YPLL and SEYLL may be an added value.

We appreciate your comment and suggestion and agree with your point. Other than the definition, formula, and separate results for YPLL and SEYLL.

 We have edited "Table 2: Characteristics of selected studies" by separating the rows for YPLL and SEYLL papers. 

 We also included this statement in the first paragraph of the discussion to highlight the contribution of this study;

 (Page 19, line 173-174) “To our knowledge, this is the first literature review examining CVD mortality using difference the method of YLL.. “

 (page 19, line 178-180) “This reviewer, extends the existing literature by reporting the year of life lost methods for premature CVD mortality using observed data…”

2 Why did the authors include non-nationally representative studies in their evidence synthesis? Studies without a specified data source, or those with a cohort design may have biased results. The premature mortality of a region should not be inferred from such studies. Furthermore, the findings from Fene 2020 that used GBD 2017 cannot be compared with other studies that are based on vital registration data. Data source of the studies should be strongly considered for correct evidence synthesis. 

Thank you for your comment. We take note of your question regarding non-nationally representative studies in the evident synthesis. Most studies used the registry data that represents the country's burden, for example, in treating, screening, and all public health programs under the umbrella of health services. Therefore, they will include all populations that were registered under their umbrella. It is very difficult to distinguish between national and non-national populations. Therefore, after discussing with all authors, we decided to remove studies that included non-nationally representative samples from our manuscript. As a result, we removed the Fene 2020 study and five cohort studies and re-analyzed all of the results. All tables, figures (including Tables 2-3 and Figures 2–7) and sentences (Result and discussion) were updated with current data.

3 There are not enough data points for assessing the trend of premature mortality in each region or in the world. If the authors utilized findings from the GBD study, it would make sense to present the trend. 

Thank you for comment. We stand to differ. If we analyze observed data, then we present the pattern/trend. We believe it is inappropriate to include data from the GBD study in our trend analysis. This is because results from GBD are estimated (predicted results) rather than observed results (reference 1). 

Therefore, we added the statement about limited data points for assessing the trend (page 20, line 256-257) and excluded data from GBD studies (page 20, line 258-260) to our limitation.

Reference 1: 

Kristin Voigt, Nicholas B King, Out of Alignment? Limitations of the Global Burden of Disease in Assessing the Allocation of Global Health Aid, Public Health Ethics, Volume 10, Issue 3, November 2017, Pages 244–256, https://doi.org/10.1093/phe/phx012

4 The first paragraph of the results section has redundant data with Figure 1 and should be summarized. 

Thank you very much for your comments. The first paragraph of the results section has been edited [Page 12, Line 3-11].

5 The conclusion paragraph should be presented as the 'Implication' paragraph in the discussion section. The conclusion section should be the gist of the manuscript, with limited discussion of the findings or their implications. 

We appreciate your comment. The conclusion was revised as you suggested (page 21, line 268-276).

6 In contrast with 'Gender', 'Sex' is biologically determined, and it is recommended to use 'Sex' instead of 'Gender' in scientific texts 

Thank you so much for your suggestion. We agree with you. The word gender has been changed to sex throughout the text.

Response to reviewer’s 2 comments

Dear Editor, 

Many thanks for this precious opportunity you gave me to review this paper.

The authors made a systematic review on literatures to evaluate the years of potential life lost (YPLL) and standard expected years of life lost (SEYLL) of premature cardiovascular disease.

The study is well conducted and results presentation is satisfactory, however, it should be interpreted in the context of several possible limitations which could be overcome with a major revision:

Author’s response

Dear reviewer, we greatly appreciate your feedback. We have provided our point-by-point response to each of your comments below.

1 Authors should state the “Premature CVD death” cut-off. It usually considers as 45/55 in females/males, but it varies from 45-65 in literatures. Although the authors stated the definition of premature mortality in different studies on “Results, Line 40”, but the main definition in the current study should be clear. 

We appreciate your comment and agree with your point. We already included the definition for study eligibility under Methodology < study selection (Page 4, Line 159-163) as below; 

“For YPLL (formula by Gardner or Romeder), any upper age limit (e.g., < 70 or < 65) that was defined by the study as premature mortality was included in this review. While the formula from GBD, usually used term SEYLL, needed standard expected years of life for each age group. Therefore, any standard life expectancy used by studies to calculate SEYLL was accepted.”

2 Authors assessed the pattern of years of life lost from CVD, which I believe that it includes a thoughtful error in the term of generalization. In each decades the available studies are different. This trend could not be explained by the total number of events, it would be due to the studies region in each decade. To be clearer, YPLL in 1990-2000, measured from one study, which conducted in Europe. These results could not be declared as the “trend of premature CVDs”. 

Thank you for your comment. We changed the sentences based on your suggestions. We also re-analyze the data and represented the data in the way that are suggested by you (in point 3). 

3 To evaluate the trend, I suggest to add a table which have “decades” as the “columns” with “sex” and “WHO Regions” as the “rows”. Thank you for your suggestion. 

As mentioned in point 2, we re-analyzed and edited Tables 3 (summary YPLL) and 4 (summary SEYLL) based on your suggestions. However, due to a lack of data, we were unable to stratify the study time by region and gender using a 5-decade category. As a result, we classified study time (decade) into two categories (before and after year 2000).

4 In the discussion, the key risk factors in LMICs should be noted. Also, mentioning the previous beneficial policies to reduce and prevent the premature CVD deaths in other countries should be examined. 

Appreciate your suggestion, we added the sentence with citation about key risk factors in LMIC (page 20, line 226-228) and previous beneficial policies to reduce premature mortality (page 20, line 228-229).

Minor comments 

1 Introduction: to evaluate the trend of NCDs it would be better to use the most recent GBD studies. Thus, I suggest to replace the reference 2 with the GBD 2019 study. 

We appreciated your suggestion. We changed the references as suggested. For your information, there are some modifications to the sentence's introduction (paragraph 1). We rephrased the introduction as suggested by reviewer 4 (suggesting the introduction to be more concise).

2 In Introduction, line 133 SEYL should change to SEYLL 

Thank you for your comment, the word SEYL has been changed to SEYLL

3 I believe the third paragraph of introduction is more about the method and Statistical analysis and should not be presented as introduction. 

Thank you for the suggestion. The authors feel that explaining the measures of YPLL and SYELL is vital and shall be retained in the text to enhance reader’s understanding. We hope the reviewer will consider our decision to keep this statement in the introduction.

4 Authors claimed that they aimed to determine global temporal trends in premature CVD mortality, I’m afraid this paper could not reach this goal. 

Thank you for your comment. We had removed the aim of the study, "to determine temporal trends in premature CVD mortality," in the last paragraph of the introduction. This statement is also added to our limitation (page 20, line 248-252);

"There is missing information from some (state how many) countries, thereby limiting global representativeness. However, as with other meta-analysis, it is always impossible to obtain complete data that are relevant to the objective of the analysis"

Response to reviewer’s 3 comments

In this manuscript, Hasani and colleagues conducted a systematic review to assess years of life lost due to cardiovascular disease by analyzing studies reporting cardiovascular disease-related premature mortality.

This is, as far as I am aware, the first systematic review of this topic (except for the GBD study).

The manuscript, however, cannot be accepted in its current form and requires major revision.

Author’s response

Dear reviewer, we are very grateful for your comments. Herewith we represent our point-by-point response to each comment made by you.

Major Comments: 

1 The search strategy might not be inclusive and requires some revision. I suggest that the authors also include non-mesh terms with OR in the first group of keywords in their PubMed search. Some examples of the keywords that can be added are: "Cardiovascular Disease"[Title/Abstract] OR “Heart Disease” [Title/Abstract] OR Aneurysm[Title/Abstract] OR Cardiomyopath*[Title/Abstract] etc. The authors can retrieve and add all of the relevant keywords from the MESH tree. 

Thank you for your comment, we had changed the keyword search as suggested by you. We updated the rescreening result on Figure 1: Flow Diagram of the Selection Articles. Also, in "Supplement 1 – search term," the search strategy was updated.

Moreover, I cannot understand why the authors did not choose to search title/abstract for their second group of keywords in PubMed instead of just searching in the titles. I also suggest that the authors avoid using filters on PubMed as it may lead to missing some relevant studies.

Thank you very much for your comment. We do agree with you. We followed Cochrane hand-book of systematic review and we adapted the method for PubMed search and search in other electronic databases. Our standard in the databases search is according to Cochrane hand book. We have search title/abstract in the group of the keywords. However, we want be very specific when we search premature CVD mortality in view of search of title/abstract resulted a lot of studies been duplicate an irrelevant.

To be clearer, numerous unrelated documents were returned when we used Title/Abstract in which the authors justified the burden of disease associated with their topic by referring to “premature mortality” in the introduction (abstract). In fact, their studies did not measure premature mortality.

2 The authors stated, "We did not include reviews, meta-analyses, letters, comments, or editorials. "Nevertheless, these study types may also contain pertinent data (please see https://doi.org/10.1161/circ.132.suppl 3.17368 for more information on heart failure). I propose that the authors reconsider their decision. 

Thank you for comment. Following the Cochrane handbook and systematic review, the methodology for writing systematic reviews only includes primary paper. We do not include any reviews and meta-analyses. However, we cross-reference the studies included in systematic reviews and meta-analyses and select the primary studies that are relevant to ours.

For letters, comments, and editorials, we do not include them in our systematic review, because it does not fit our eligibility criteria as stated in Cochran hand book

About this paper; https://doi.org/10.1161/circ.132.suppl 3.17368

We apologize, as we could not find the article as you suggested.

3 In the Prospero protocol, the authors indicated that they would also search Embase and CINAHL. Please include these databases or indicate that you were unable to adhere to the primary protocol.

Thank you for your comment. We included EMBASE and CINAHL as reported by Cochrance Library. CINAHL and EMBASE are both part of the Cochrane Library. However, we could not find any studies in this database. 

4 In the results section, the authors have stated in multiple sentences whether the trend has increased or decreased. This study was unable to demonstrate statistical significance for any of these trends 

We appreciate your comment and agree with your point. There is a Mann-Kendall trend test. But we need to have YPLL/SEYLL for each year. In our case, this is probably not appropriate, as we have a few studies in a year and none in the next year.

Mann-Kendall trend test - https://www.statology.org/mann-kendall-trend-test-r/

However, as suggested by reviewer 2, we re-analyzed and edited Tables 3 (summary YPLL) and 4 (summary SEYLL), adding the stratification analysis of study time by region and sex. We also revised the sentences about the trend.

In addition, we added this sentence in the first paragraph of discussion “Even though this review could not measure the magnitude of the change in trend, our descriptive findings identified a slight reduction…” (page 19, Line 187-190)

5 In the discussion, please consider including a section on the potential impact of the COVID-19 pandemic on the burden of premature mortality attributable to cardiovascular diseases. Please investigate whether any studies exist that compare the burden prior to and during the pandemic. 

Thank you for your valuable suggestion. We added a one paragraph regarding the impact of the COVID-19 pandemic on premature CVD mortality and a suggestion for further studies (page 20, Line 235-244).

6 The discussion requires further elaboration on preventive measures and efforts that can improve the quality of care in high-income and low-income countries. 

Appreciate your suggestion, a few sentences with citation were added in page 20 line 229-233.

Minor comments: 

1 Could the authors include the extraction sheet as supplemental material to increase the study's reproducibility? Thank you for your advice.

We will upload the extraction sheet form as Supplement file (Supplement 1). 

During submission, we also share the link (as shown below) for data and r code. If this paper is accepted and published, the link will be included in the "Data Availability Statement" section of the manuscript.

https://github.com/shakirarodzlan/SR_PrematureMortality.git

2 Since few studies were conducted to assess the burden of premature mortality due to HF, I questioned whether the authors attempted to obtain the necessary information from the following paper. http://dx.doi.org/10.1136/heartjnl-2020-317833

Thank you for your suggestion. This paper ("Association of heart failure and its comorbidities with loss of life expectancy") is already included in our screening process and eligible for full text review. However, after full text review, we exclude this paper for several reasons, as below. On Figure 1, we label this paper as "report excluded—not presented exact YLL value."

Second screening (full text) 

- Respondents among patients with heart failure

- Not direct measure premature CVD mortality

- Calculated excess mortality and excess loss of life 

- Not presented exact value for YLL

3 Please do not use the abbreviated form of "it's" on page 18, line 166. 

We appreciate your comments. We changed “it’s” with appropriate word. Kindly refer to page 16 line 135 (edited version).

4 Please note, on page 19, line 197, that according to the GBD study, the rate of All ages global YLL due to cardiovascular diseases did not differ significantly between 2019 and 1990 (95% UI= -0.12 to 0.01 (GBD Results tool)) 

Appreciate your comment and apology for overlooking this matter. We rephrased the sentence in accordance with our finding and citation (refer to page 19 line 190-192). 

We cited this study;

Masaebi F, Salehi M, Kazemi M, Vahabi N, Azizmohammad Looha M, Zayeri F. Trend analysis of disability adjusted life years due to cardiovascular diseases: results from the global burden of disease study 2019. BMC Public Health. 2021;21: 1268. doi:10.1186/s12889-021-11348-w

5 In figure 4, why did the authors choose these countries as opposed to all? Please add the number of available studies from each country to the respective bar. 

Thank you for your valuable comments and insight. Figures 4 (and 3) show all countries with available data that provided values for the YPLL and SEYLL rates. We apologies for misunderstanding the caption. Thus, we have revised the title for Figure 4 (and Figure 3 as well) for better clarity. We changed the caption to "The SEYLL rate of countries with available data."

Response to reviewer’s 4 comments

This is a well conducted and well reported systematic review focusing on YLLs of CVD to determine the burden of premature CVD mortality on a global scale.

I have some comments and questions:

Author’s response

Dear reviewer, we greatly appreciate your feedback. We have provided our point-by-point response to each of your comments below.

1 The main issue is to determine how representative these data are of the population in that region/time? I suppose most of the included publications are not done with nationally representative data. Even if they are national, it would be a big assumption to take these estimates to represent a whole WHO region. I think the paper should put less emphasis on the regional results. 

Thank you for your comment. We take note of your question regarding representative studies in the evident synthesis. We agree that data from some countries might have limited quality and representativeness. Most studies used registry data to represent the country's burden, such as in treating, screening, and all public health programs that fall under the umbrella of health services. They will include all populations that were registered under their umbrella. It is very difficult to distinguish between national and non-national populations. Therefore, after discussing with all authors, we decided to remove studies that included non-nationally representative samples from our manuscript. As a result, we removed the Fene 2020 (GBD study – estimated data) and five cohort studies and re-analyzed all of the results. For our analysis, we assumed that all data in this study are valid, and they represent the national data. We added this statement under method (in page 6 Line 211-213):

“In the analysis, we exerted two additional assumptions: a) We assumed that data from each source represents the national population and b) the measurement of the data were valid for all data sources”.

2 The GBD study provides a great source, which many researchers use. The authors can discuss the advantages and disadvantages of their study, which I understand uses real-world data without modeling, to the GBD estimates, considering that GBD study provides a much more comprehensive picture of the CVD YLLs. We appreciated your comment. On the first paragraph of the discussion, we added the sentence about the strength and limitation of GBD study and the added value of our review (page 19 Line 174-180); “Although GBD is the most widespread global estimate…”

3 Median and range are reported for YLLs, yet it is not clear how these were calculated and reported. I am not sure how reporting median/range of included studies’ results, especially in groups with 3 studies or even 1 study, is useful. Moreover, the range for the total number of studies is so wide, it does not convey a clear message. Perhaps the authors can clarify their methods and discuss these limitations. 

Thank you for your valuable comments.

We presented median and range for the YLLs to summaries our result. We added the details explanation about median calculation in the method (page 6 Line 202-208). We also added this statement as a footnote to Table 3 (and Table 4 as well) to help the reader understand;

“Each study reported either the YPLL rate or the YPLL per death or both. They may provide YPLL values for each CVD type and sex or for the overall population. We treat each value as separate data for each paper to calculate the median and IQR (Interquartile range)”.

The wide range of the YLLs reflected the heterogeneity of the studies, which is something that we think the readers should be aware of. Possible cause of the heterogeneity included age coverage of the participants and methods used in each individual study. We have acknowledged this in the limitation section (Page 20, line 260-264). We hope we have addressed these issues adequately. 

4 Figure 5 shows a world map of SEYLL rates. I think such a figure has a limitation in that these estimates are not from the same time period, but rather come from studies over five decades. 

Thank you for your suggestion. We agree with your point. We analyzed our data and removed Figure 5 because it was irrelevant to our current analysis.

We also added this statement to our limitations: "Some studies are from different time periods. As a result, we were unable to identify a global study conducted during the same time period” (page 20, line 252-253).

5 One of the included studies, from the Caribbean countries, has GBD study 2017 as its source data. Since GBD studies only provide estimates, I think it would be better to exclude this record. Or perhaps there was some reason behind including this study that the authors can clarify. 

Thank you for your comment. We agree that GBD results are estimated results so adding them to our analysis might not be completely right. Thus, we decided to exclude Fene 2020 study (Carribean countries) and re-analyze all results. All tables and figures (including Tables 2-3 and Figures 2–7) and sentences (Result and discussion) were updated with current data.

Minor comments: 

1 In figure 1, among the reasons for exclusion of full-texts, it seems reporting only absolute YLL is written twice. Please recheck it 

We appreciate your comment. The sentence was edited. Kindly refer to the updated Figure 1.

2 I think the introduction can be more concise. This would leave room for a more complete discussion of the findings, and what the study adds to the literature. 

Thank you for the suggestion, first paragraph has been edited to highlight CVD statistics in justifying the conduct of this systematic review (page 3, Line 86-89). The authors feel that explaining the measures of YLL and SYELL (in subsequent paragrahs) is vital and shall be retained in the text to enhance reader’s understanding.

---

## [Decision Letter · Decision Letter 1]

20 Mar 2023

The burden of premature mortality from cardiovascular diseases: a systematic review of years of life lost

PONE-D-22-34213R1

Dear Dr. Hasani,

We’re pleased to inform you that your manuscript has been judged scientifically suitable for publication and will be formally accepted for publication once it meets all outstanding technical requirements.

Kind regards,

Sina Azadnajafabad, MD, MPH

Academic Editor

PLOS ONE

Additional Editor Comments (optional):

Reviewers' comments:

Reviewer's Responses to Questions

**Comments to the Author**

1. If the authors have adequately addressed your comments raised in a previous round of review and you feel that this manuscript is now acceptable for publication, you may indicate that here to bypass the “Comments to the Author” section, enter your conflict of interest statement in the “Confidential to Editor” section, and submit your "Accept" recommendation.

Reviewer #1: All comments have been addressed

Reviewer #2: All comments have been addressed

Reviewer #3: All comments have been addressed

Reviewer #4: All comments have been addressed

2. Is the manuscript technically sound, and do the data support the conclusions?

Reviewer #1: Partly

Reviewer #2: Yes

Reviewer #3: Partly

Reviewer #4: Yes

3. Has the statistical analysis been performed appropriately and rigorously? 

Reviewer #1: N/A

Reviewer #2: Yes

Reviewer #3: Yes

Reviewer #4: I Don't Know

4. Have the authors made all data underlying the findings in their manuscript fully available?

Reviewer #1: Yes

Reviewer #2: Yes

Reviewer #3: Yes

Reviewer #4: No

5. Is the manuscript presented in an intelligible fashion and written in standard English?

Reviewer #1: No

Reviewer #2: Yes

Reviewer #3: Yes

Reviewer #4: Yes

6. Review Comments to the Author

Reviewer #1: (No Response)

Reviewer #2: (No Response)

Reviewer #3: I thank the authors for their detailed response. Most of the comments have been addressed. While the authors made a valid point by their response to the second comment, I still believe there might be some studies that have been missed. For instance, they did not try to include a relevant study that I had previously suggested. It is important for a systematic review to detect and include all of the eligible studies.

Reviewer #4: I thank the authors for their thoughtful and thorough response. After a careful evaluation of the manuscript, I think this paper may have some limitations, but this is the nature of all scientific investigations. Overall, it is a worthy contribution to this topic. In my opinion, the manuscript is acceptable.

7. PLOS authors have the option to publish the peer review history of their article (what does this mean?). If published, this will include your full peer review and any attached files.

Reviewer #1: No

Reviewer #2: No

Reviewer #3: No

Reviewer #4: No

---

## [Editor Report · Acceptance letter]

13 Apr 2023

PONE-D-22-34213R1 

The burden of premature mortality from cardiovascular diseases: a systematic review of years of life lost 

Dear Dr. Rodzlan Hasani:

I'm pleased to inform you that your manuscript has been deemed suitable for publication in PLOS ONE. Congratulations! Your manuscript is now with our production department. 

Kind regards, 

on behalf of

Dr. Sina Azadnajafabad 

Academic Editor

PLOS ONE